# Patients' experiences with musculoskeletal spinal pain: A qualitative systematic review protocol

**Alaa El Chamaa** [ORCID]*, **Katie Kowalski** [ORCID], **Pulak Parikh, Alison Rushton** [ORCID]

Faculty of Health Sciences, Western University, London, Ontario, Canada

* aelchama@uwo.ca

**Data Availability Statement:** No datasets were generated or analysed during the current study. All relevant data from this study will be made available upon study completion.

## Abstract

### Background

Musculoskeletal (MSK) spinal pain encapsulates various conditions including lumbar (low back), cervical (neck), and thoracic pain that significantly impact individual and global health. While clinical aspects of spinal pain have been well-studied, understanding patients' personal narratives and lived experiences remains essential for enhancing patient-centered care, improving treatment adherence, and informing healthcare policies. It provides deep insights into the impacts of spinal pain, guiding more effective and empathetic treatment approaches. This systematic review aims to synthesize qualitative evidence on patients' experiences with MSK spinal pain, providing insight into the challenges faced, coping strategies, daily life impacts, and healthcare interactions. The objective of this review is to synthesize the qualitative evidence regarding the lived experiences of patients with MSK spinal pain.

### Methods

This systematic review will use a meta-aggregation approach to synthesize data from qualitative studies, that will be identified through a comprehensive search of electronic databases and supplemented by grey literature searches. Two independent reviewers will screen, identify, and extract data from eligible studies. In cases of disagreement, conflicts will be resolved by consulting a third reviewer. These same reviewers will then use the Joanna Briggs Institute (JBI) qualitative quality assessment tool to evaluate the methodological quality of the identified studies, with the derived scores informing the synthesis process, that will involve extracting each study's findings along with their supporting illustrations, then grouped into categories based on similarity in meaning. These categories will then be aggregated to form synthesized findings.

### Implications

Synthesized findings on patients' lived experiences with MSK spinal pain including key themes, patterns, and insights will be presented. By emphasizing patient narratives, the

**Funding:** The author(s) received no specific funding for this work.

**Competing interests:** The authors have declared that no competing interests exist.

results of the review can contribute to the optimization of outcomes, and to enhance patient-provider relations and improve quality of care in MSK spinal health.

## Introduction

Musculoskeletal (MSK) spinal pain encapsulates various conditions including lumbar pain (low back pain (LBP)), cervical (neck) pain, and thoracic pain; significantly impacting global health. Low back pain is the most common, with a prevalence of 619 million people affected worldwide in 2020, and projections suggest an increase to 843 million by 2050 [1]. LBP is not only prevalent [2–5], but is also a leading cause of disability, imposing substantial economic and healthcare burdens [6–8]. According to a preprint article, neck pain is also a very common health condition with a prevalence of 203 million people affected in 2020, projected to rise to 269 million by 2050 [9]. Neck pain ranks among the top contributors to global disability, also leading to substantial economic burdens [10, 11].Thoracic pain, while less prevalent than LBP and neck pain, poses a significant challenge to those affected. Though not as widespread, thoracic pain contributes notably to disability, and incurs considerable economic burdens comparable to those of more common spinal pain conditions [12, 13]. It is less researched, and often overlooked [13]. Beyond the clinical and physiological exploration, there is a rich tapestry of personal narratives of experiences, emotions, and beliefs that people with MSK spinal pain present with daily. While there is a substantial body of quantitative research focusing on the clinical and epidemiological aspects of spinal pain, qualitative investigations exploring the lived experiences of patients are comparatively scarce. After decades of researching and studying the spine, only 38 unique qualitative studies addressing patients' experiences with spinal pain (neck and LBP) were included in previous systematic reviews [14–16]. This gap indicates a significant underrepresentation of patient-centered perspectives in the existing literature, highlighting the need for more qualitative research. The significance of understanding patients' experiences with spinal pain cannot be understated. Currently, patients "struggle to be seen and understood as a person", and they have "a desire to be taken care of and listened to" [17]. Understanding their experiences can help shed light on patients' daily challenges, their coping strategies, and can influence healthcare interactions, compliance with treatments, and the overall quality of life [18]. Insights into the lived experience of people with MSK spinal pain can guide healthcare providers in tailoring patient centered care which will optimize treatment outcomes and enhance patient-provider relations [18].

Qualitative research is the research method that can "make the world visible" "attempting to make sense of, or to interpret, phenomena in terms of the meanings people bring to them" [19]. Qualitative research enables exploration of the narratives and phenomenological experiences of patients with MSK spinal pain. A systematic review can collate, evaluate and synthesize qualitative findings on a broader scale, providing an understanding of patients' experiences, highlighting common themes, disparities, and gaps or areas that need further exploration. To identify potential gaps and weaknesses in the current body of research, we used the AMSTAR 2 ('A Measurement Tool to Assess systematic Reviews') tool to rate the overall confidence in the results [20] of previous systematic reviews. It has 16 items with yes and no answers and an overall rating that could be high (none or one non-critical weakness), moderate (more than one non-critical weakness), low (at least one critical flaw) and critically low (more than one critical flaw) [20].

Three previous systematic reviews of qualitative evidence have synthesized patients' experiences with LBP and neck pain [14–16]. The first was a critical review of qualitative studies that explored the perceptions and experiences of individuals with neck pain, in which they included 9 qualitative papers and Identified themes related to the impact of neck pain on individuals' lives, including "Physical theme: 'My neck has gone wrong'", "Psychological theme: 'I am worried about my recovery'", and "Social theme: 'Pain limits my life'" [14] (AMSTAR 2 rating: low overall confidence in the results of the review). The second systematic review Explored the experiences of individuals living with chronic low-back pain (CLBP) and included 25 papers from 18 studies, key findings were "social construction of CLBP", "psychosocial impact of CLBP", "coping with CLBP", "concept of biographical suspension: suspended 'wellness', 'self', and 'future'" [15] (AMSTAR 2 rating: moderate overall confidence in the results of the review). The third systematic review was a meta-ethnography (a qualitative synthesis method in which the reviewer re-interprets the data from the original studies [21]) of qualitative research on patient experiences of chronic LBP This review included 38 qualitative studies with key findings of "undermining influence of pain", "disempowering impact on all levels", "unsatisfying relationships with health care professionals", and "learning to live with the pain" [16] (AMSTAR 2 rating: moderate overall confidence in the results of the review). No systematic review of qualitative studies has explored patients' experiences with thoracic pain, which aligns with the previous statement that this region of the spine is less researched and often overlooked [13].

Several important gaps in the literature have emerged from previous systematic reviews of qualitative evidence exploring the lived experience of MSK spinal pain. Firstly, the absence of systematic reviews that specifically synthesize qualitative evidence on thoracic pain is a critical gap in the existing literature, unlike low back pain and neck pain, which have been the focus of previous reviews. Second, previous reviews focused narrowly on a specific region of spinal pain, highlighting the lack of a comprehensive synthesis and a clear need for a more inclusive review that consolidates findings of spinal pain across the different spinal regions. By synthesizing data from studies across all spinal regions, our comprehensive review can improve the overall methodological quality. A broader scope allows for a more extensive and diverse set of studies, enhancing the robustness and reliability of our findings. This comprehensive approach also enables us to identify overarching themes and unique differences in patients' experiences across different spinal regions, providing a more nuanced understanding that single-region reviews might miss. Furthermore, understanding the full spectrum of spinal pain experiences across different regions can inform more integrated patient-centered care strategies, aligning with the principles of holistic healthcare. A third gap in the literature is the moderate to low confidence in the findings of previous systematic reviews, which reflects some methodological limitations and challenges in fully capturing the complexity of patients' experiences. Previous reviews used quality assessment tools biased toward certain paradigms or methodologies of qualitative research. In contrast, we will be using a more balanced and comprehensive assessment of qualitative studies, ensuring methodological quality and reducing bias [22]. Additionally, by expanding our search to include all spinal regions, we ensure a more thorough and inclusive literature search strategy, reducing the risk of missing relevant studies. Our review will also provide a detailed description of the characteristics of included studies, including the context and specific spinal regions involved, enhancing the understanding of applicability and transferability of the findings. In summary, this qualitative systematic review will address these gaps in the literature by encompassing a wide range of qualitative studies for a comprehensive synthesis of findings on patients' experiences with spinal pain across various regions (low back, neck and thoracic) and employing a robust methodological framework that will result in higher confidence in the findings.

## Objective

The objective of this review is to synthesize the qualitative evidence regarding the lived experiences of patients with MSK spinal pain, emphasizing a dual approach.

Firstly, we will conduct three individual syntheses for each of the spinal region (lumbar, cervical, and thoracic) to understand the unique challenges and experiences associated with each area. Secondly, we will perform a more comprehensive synthesis that will combine findings across the different regions. This approach will allow us to identify both region-specific patterns and some broader, overarching themes for MSK spinal pain. By doing so, and extending the scope to incorporate thoracic pain, this review aims to address a significant gap in the existing literature and consolidate evidence for a broader comprehensive understanding, offering healthcare providers a holistic view of the patients' journey and allowing for cross-comparative insights, richer data and enhanced generalizability. The review will collate and evaluate qualitative findings to gain an understanding of the challenges, coping strategies, impacts on daily life and healthcare interactions. This comprehensive synthesis could contribute to enhancing patient-centered care approaches and inform future directions in the field of MSK spinal pain management.

## Methods

### Design

This qualitative systematic review protocol is reported according to the established PRISMA -P (Preferred Reporting Items for Systematic review and Meta-Analysis Protocols) statement [23] and prospectively registered in PROSPERO (International prospective register of systematic reviews) to enhance transparency and prevent duplication of research efforts [24], registration ID: CRD42024467359. The review team consists of physical therapy researchers with expertise in qualitative research, systematic reviews and spinal pain.

### Ethical considerations

This systematic review protocol was designed to synthesize existing qualitative literature on musculoskeletal spinal pain experiences. As it does not entail direct human participation, patient data collection or animal experimentation, but rather compiles and analyzes previously published data, it does not fall within the purview of ethics review requirements. The protocol adheres to all applicable guidelines and standards for ethical research conduct in the compilation and synthesis of secondary data.

### Patient and public involvement

We engaged with a patient partner advisory group at Western University, comprised of individuals who have personal experiences with MSK spinal pain. The engagement was structured as a presentation of the proposed review protocol followed by a discussion where the group thought that the project is "extremely important" and "very impactful", which affirmed the relevance and necessity of this review. Furthermore, the group said that the proposed review protocol was "very clear" and did not recommend any changes to its design, further validating our approach in this study.

### Eligibility criteria

The SPIDER tool has been used to guide eligibility criteria as it is more appropriate for qualitative systematic reviews [25].

- S (sample): adults (18 years and above).

- PI (phenomenon of interest): MSK spinal pain (neck, thoracic and low back). Studies involving patients with non-MSK spinal pain (e.g. spinal cord injury, cancer, etc.) will be ineligible.

- D (design): Any qualitative research study design (e.g. interviews (structured, semi-structured), observational, focus groups, etc.)

- E (evaluation): Patients' experiences, feelings, beliefs, understandings, etc.

- R (research type): Qualitative and qualitative data from mixed methods research.

We will be including mixed methods studies, but we are specifically interested in their qualitative data, and the quantitative aspects will not be used or included in our analysis. No limits will be set for publication year or language. For potentially eligible studies published in languages other than English, if none of the review team are proficient in that language, the corresponding author will be contacted to determine if an English version is available. If unsuccessful to contact the authors or no English version is available, an open-source software (Google Translate and ChatGPT (Chat Generative Pre-Trained Transformer) artificial intelligence chatbot) will be used to translate the texts. To ensure that the translations are valid, a bilingual individual familiar with the subject matter will verify the translations and confirm the accuracy and context of the translated material.

## Information sources

Databases (Medline (via OVID), CINAHL, Scopus, Web of Science, and EMBASE) will be systematically searched using a combination of keywords and Subject headings (Mesh, EMTREE, CINAHL Headings, etc.) The grey literature will also be searched using specific databases (OAIster, OpenGrey, and ProQuest Dissertations) and Google searches employing the same search terms and importing the first 10 pages. As it is not possible to screen all the results from Google, we will be relying on the engine's "power of relevancy that brings the most relevant results to the top of the list" [26].

## Search strategy

A comprehensive search strategy was developed in consultation with a qualified librarian at the University of Western Ontario, exemplified in S1 Table. The strategy was developed around the constructs of musculoskeletal spinal pain and qualitative and mixed methods studies. Search terms for the MSK spinal pain construct were informed by the National Institute for Health and Care Excellence guidelines for low back pain and sciatica in over 16s [27] and the Clinical Practice Guidelines Linked to the International Classification of Functioning, Disability and Health from the Orthopedic Section of the American Physical Therapy Association for Neck Pain [28]. Additionally, CADTH (Canadian Agency for Drugs and Technologies in Health) search filters will be applied to databases to filter for qualitative and mixed methods studies [29]. The complete search strategy will be adapted for each database and will be included in the final review manuscript. See S1 Table.

## Study records

**Data management.** Covidence will be used to manage data for transparency and to facilitate collaboration between the review team members in multiple review phases (i.e. title and abstract screening, full text screening and data extraction) [30].

**Selection process.** Initially, two independent reviewers, (A.E. and R.B.) will conduct the title and abstract screening of all identified studies from databases (Medline, CINAHL, Web of Science, and EMBASE) as well as results from the grey literature search (OAIster, OpenGrey, ProQuest Dissertations, and Google searches) based on the predetermined inclusion and exclusion criteria. Studies that potentially meet criteria will undergo full-text screening for the final inclusion decision in the review. Throughout this process, any discrepancies at any stage will be resolved through discussion first, if no agreement can be reached, a third reviewer (AR) will be consulted to make the final decision.

**Data collection process.** Two independent reviewers will use a data extraction form that will be developed for the purpose of this review, as detailed in S2 Table. They will be trained in the data extraction form, which will be piloted before being used and may be modified as required. Any discrepancies will be resolved through discussion, if no agreement can be reached, a third reviewer will be consulted to resolve it. The following data will be extracted from the included studies: study characteristics (publication year and geographical location of the study), sample information (size, participant demographics, inclusion/exclusion criteria), study aims, study design (methodological underpinnings), data collection methods, key findings which would be the author's verbatim analytical interpretation of the result or data [31, 32] accompanied with an illustration that informs this finding, and that may be verbatim quotes or narratives from the participants, observations or any other supporting data [31]. Extracting first and second order constructs [32, 33] will give us the opportunity to engage with the raw data (mainly participants' quotes) alongside the researchers' interpretations of these quotes and ensures that review findings are closely tied to the original experiences of the participants [32].

## Outcomes and prioritization

The primary outcome of this review is to identify and synthesize key themes and patterns of the lived experiences of patients with MSK spinal pain, which will include exploring personal narratives, coping strategies, challenges and interactions with healthcare system. While the secondary outcome will focus on identifying gaps in the current literature. The review will prioritize outcomes that provide the most comprehensive insight into the lived experiences of MSK spinal pain patients.

## Quality assessment

Unlike quantitative research, where quality criteria are known and agreed on [34], qualitative studies are more complicated to appraise for quality. There is still no gold standard for assessing quality and there have been many debates over the years about the subject of criticizing qualitative research [34–41]. Some scholars even argue that it is not possible to appraise qualitative work using a common framework (gold standard) and looking for one should be abandoned [42]. Some tools have been developed for the purpose of assessing the quality of qualitative research, such as the critical appraisal skills program (CASP), the evaluation tool for qualitative studies (ETQS), and the Joanna Briggs Institute (JBI) tool. In this review the JBI tool will be used for the quality assessment of the identified studies, as it focuses on congruity [43] to evaluate the quality of the research while accommodating for the complexity and the diversity of different paradigms and methodological approaches. This is in contrast to the other tools that have a more fixed and rigid set of appraisal criteria, that cannot fully address the congruence between the research question, methodology and methods to the extent that the JBI tool does [22]. This makes the JBI tool particularly more valuable as it offers a more flexible and inclusive approach in assessing quality. The two reviewers will use the tool to

assess methodological quality of each paper independently. Any discrepancies will be resolved through discussion If no agreement can be reached, the third reviewer will be consulted. The reviewers will initially be trained on the JBI quality assessment tool. They will pilot it to appraise 3 articles, meet to discuss consistency in application of the tool, and then continue with the rest of the assessment.

The JBI tool consists of ten questions with possible answers of Yes, No, Unclear or Not applicable [31]. A decision to include or exclude a study from the synthesis based on a predetermined cut off percentage score, as is suggested by JBI [31], will not be used in our review. Given the specialized nature of the review and the scarcity of the research, excluding studies based on a quality threshold might further narrow an already limited pool of research. We are also trying to capture a broad spectrum of perspectives, as each qualitative study offers some unique insights into patients' experiences. By not excluding any relevant studies we will be preserving these comprehensive insights, enhancing generalizability and avoiding some potential selection bias that could influence the synthesis [32]. Instead, we will be giving each study a percentage score based on the proportion of "Yes" answers while weighing all items equally. This assessment will help provide an insight into the methodological strengths and limitations of the identified studies. The score will play a crucial role in the meta-aggregation synthesis process, as it will be used to inform the interpretation of findings from individual studies, the ones with higher scores will be given more consideration and higher plausibility levels in the synthesis due to their methodological quality [31]. The scores will also guide the weighting of each study's contribution to the meta-aggregation categories to ensure that the synthesis is balanced and not influenced by studies with methodological limitations [31]. By transparently reporting quality scores, we will also be enhancing the credibility of our review, allowing readers to appreciate the robustness of the evidence base. Furthermore, by assigning lower priority to lower quality studies, we mitigate their potential negative impact on the overall synthesis. This means that while all studies are included to preserve the breadth of perspectives, the conclusions drawn are primarily informed by higher quality evidence. This approach maintains the integrity of our findings, as the influence of studies with methodological limitations is proportionately reduced, ensuring robust and credible synthesized results. For mixed methods studies, quality assessment will focus solely on the qualitative components using the same JBI tool for qualitative studies, this ensures that we assess the methodological quality relevant to our review's qualitative focus.

## Data synthesis

We recognize and acknowledge the issues with synthesizing qualitative data, as researchers explained that some qualitative data synthesis methods like meta synthesis allows for a deep, interpretative synthesis that can generate new theoretical insights and understanding, providing a rich and nuanced comprehension of the research phenomenon [44]. However, it involves interpreting an interpretation, which would risk losing the core insights of the original studies [32, 33, 45, 46]. This method often involves elevating data to a more abstract level by using analytical methods like those in the initial studies (e.g. thematic analysis). While other methods like meta summary focus on compiling key findings through content analysis without the need of the same depth of reinterpretation [32, 45, 47, 48]. In our synthesis we will make sure that our findings stay closely tied to the original data, so they reflect the original participants' experiences [32], by using a meta-aggregation approach to data synthesis that does not entail a new interpretation or a new conceptualization of the evidence [31]. We will synthesize the identified themes, patterns and insights from the included studies to provide a comprehensive systematic review of the literature. We will synthesize findings from each of the three spinal

regions (low back, thoracic and neck) separately as well as an additional comprehensive synthesis of all regions together, organizing the findings into categories, exploring relations between themes and identifying gaps or variations in the literature. To achieve this, we will be following the 3 JBI meta-aggregation steps to synthesize data [31]:

1. Starting with assigning levels of plausibility to the collected key findings from included studies, based on the quality assessment score and an assessment of the degree of fit between the supporting data and the relative illustration (assessing how congruent the data is with the findings by evaluating whether the data directly supports the findings or if there are gaps or ambiguities). Each finding from the included studies can be assigned one of the three possible levels of plausibility, if the finding is supported by data from studies with good methodological quality and are beyond reasonable doubt that cannot be challenged then the finding will be given an "unequivocal" level of plausibility. If the finding is from a study with some methodological limitations and supporting illustration lacking clear associations, which could open it to challenge, then it will be given an "equivocal" plausibility level. Lastly, if a finding is from a study with serious methodological limitations and is unsupported by the data, then the finding will be given an "unsupported" level of plausibility [31]. This process will involve discussion and consensus among the research team, especially in cases where the assessment is not straightforward.

2. In the second step, we will create categories for findings. Which involves the aggregation of similar findings from different studies into categories guided by thematic similarities, with a minimum of two findings in each category to ensure that it represents a recurring theme rather than an isolated observation. Each category will represent a distinct aspect that has emerged from the data.

3. Finally, in the last step we will formulate one or more synthesized findings from at least two categories [31].

After synthesizing these findings, we will present the preliminary results to the same MSK spinal pain patient partner advisory group at Western University. This step will provide patient perspective validation, allowing the group to provide feedback on the extent to which the findings resonate with their experiences and perceptions. This iterative process can enhance the validity of our synthesis and ensures that the findings accurately reflect the lived experiences of people living with MSK spinal pain.

## Confidence in cumulative evidence

For each synthesized finding, we will be using GRADE-CERQual ('Grading of Recommendations Assessment, Development and Evaluation- Confidence in the Evidence from Reviews of Qualitative research') to assess how much confidence to place in these findings (i.e. high, moderate, low, or very low confidence) [49]. The assessment is based on four components that are: methodological limitations (will be based on the JBI tool's quality score, with higher scores indicating fewer methodological limitations [50]), coherence (assessing the data that contributed to each finding and making a judgement on how clear the fit is between the data and the finding [51].), adequacy of data (assessing the richness and quantity of data that contributed to the findings [52]) and relevance (assessing and identifying differences and similarities between the context specified in the review question and the context of the studies supporting each finding [53]) [49]. Each component will be first assessed individually and given "No or very minor concerns", "Minor concerns", "Moderate concerns" and "Serious concerns" with a description of these concerns. Findings start as "high confidence" and then are rated down by

one or two levels based on the seriousness and number of concerns for each component. The overall confidence will be rated down by at least one level for any serious concerns in any of the components. For minor or moderate concerns, the overall confidence will only be rated down if there are two or more minor or moderate concerns in one of the components [54].

## Interpretation, discussion and implications for future research and clinical practice

This systematic review will analyze findings on MSK spinal pain within the context of current research, aiming to illuminate new insights and deepen the understanding of patients' experiences, with a particular focus on patients' lived experiences and personal narratives of MSK spinal pain, examining the intricate relationship among physical symptoms, psychological impacts and social dynamics. Our focus will be to discern patterns, commonalities and disparities in these experiences, aiming to illuminate the multifaceted journey of patients dealing with MSK spinal pain.

Depth of understanding patients' lived experiences of MSK spinal pain will have important implications for future research and clinical practice. This review will not only explore the challenges faced by patients but also highlight positive aspects, such as effective coping strategies and areas where current care is successful. The discussion will delve into the methodological nuances of the included studies, emphasizing the importance of methodological rigor and suggesting improvements in study design, data collection, and analysis methods. Additionally, the systematic review may uncover gaps in the current literature, particularly concerning thoracic pain, leading to avenues for future research. This emphasizes the need for more comprehensive studies that encompass a broader spectrum of experiences with MSK spinal pain, including insights into effective patient-centered care strategies. These findings could be instrumental in informing the development of patient education and support programs, enhancing comprehensive pain management, and improving outcomes and quality of care in MSK health.

## Supporting information

**S1 Table. Search strategy medline (OVID).**
(DOCX)

**S2 Table. Data extraction form.**
(DOCX)

**S3 Table. PRISMA-P 2015 checklist.**
(DOCX)

## Acknowledgments

Western Libraries, Western University (London, Ontario, Canada) for the assistance in developing the search strategy for this systematic review.

The MSK spinal pain patient partner advisory group at Western University for their invaluable insights and contributions.

## Author Contributions

**Conceptualization:** Alaa El Chamaa, Katie Kowalski, Pulak Parikh, Alison Rushton.

**Methodology:** Alaa El Chamaa, Katie Kowalski, Pulak Parikh, Alison Rushton.

**Supervision:** Katie Kowalski, Pulak Parikh, Alison Rushton.

**Writing – original draft:** Alaa El Chamaa, Katie Kowalski, Pulak Parikh, Alison Rushton.

**Writing – review & editing:** Alaa El Chamaa, Katie Kowalski, Pulak Parikh, Alison Rushton.

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
