## [Decision Letter · Decision Letter 0]

10 Jun 2024

PONE-D-24-07892Patients’ Experiences with Musculoskeletal Spinal Pain: A Qualitative Systematic Review ProtocolPLOS ONE

Dear Dr. El Chamaa,

Thank you for submitting your manuscript to PLOS ONE. After careful consideration, we feel that it has merit but does not fully meet PLOS ONE’s publication criteria as it currently stands. Therefore, we invite you to submit a revised version of the manuscript that addresses the points raised during the review process.

We look forward to receiving your revised manuscript.

Kind regards,

Adedayo Ajidahun

Academic Editor

PLOS ONE

Reviewers' comments:

Reviewer's Responses to Questions

**Comments to the Author**

1. Does the manuscript provide a valid rationale for the proposed study, with clearly identified and justified research questions?

Reviewer #1: Yes

Reviewer #2: Yes

2. Is the protocol technically sound and planned in a manner that will lead to a meaningful outcome and allow testing the stated hypotheses?

Reviewer #1: Yes

Reviewer #2: Yes

3. Is the methodology feasible and described in sufficient detail to allow the work to be replicable?

Reviewer #1: Yes

Reviewer #2: Yes

4. Have the authors described where all data underlying the findings will be made available when the study is complete?

Reviewer #1: Yes

Reviewer #2: Yes

5. Is the manuscript presented in an intelligible fashion and written in standard English?

Reviewer #1: Yes

Reviewer #2: Yes

6. Review Comments to the Author

You may also provide optional suggestions and comments to authors that they might find helpful in planning their study.

Reviewer #1: Generally, this qualitative systematic review can contribute significantly to the literature on MSK pain management, and the description of the methods is comprehensive. However, the authors need to flesh out the uniqueness, innovative contribution, or implication of the study. The potential impact or “so what” question is not well articulated yet. The allusion to patient-centeredness is generic and has been studied by similar reviews. A quick search shows several of them.

One of your arguments on page 6 is that a comprehensive synthesis that includes neck, thoracic, and lower back regions is necessary. However, you did not provide a clear justification, in the paragraph, for that assertion. What would it add to the literature or its implication for practice?

The correct meaning of CADTH is “Canadian Agency for Drugs and Technologies in Health” (p. 10, line 187-188).

There seem to be some discordance between your sub-objective on quality decision/confidence which stems from your critique on the low quality of previous reviews (in page 6, lines 112-118) and your decision (in page 13, lines 249-252) not to exclude papers regardless of the quality rating. It is not very clear how attaching lower priority to the “lower quality” papers would steer the overall integrity of your review findings.

The manuscript needs proofreading before resubmission: punctuations and inconsistent font size.

Reviewer #2: Dear Authors

Thanks a lot for the opportunity you have offered me to revise the manuscript “". I thank the authors for their efforts in producing this study. It perfectly aligns with my research and expertise; thus, I am confident I can offer a valuable peer review.

As a significant strength, the protocol of this review synthesize the qualitative evidence regarding the lived experiences of patients with MSK spinal pain. This proposal is a novelty in the field and adds information to the existing evidence in the literature.

As a significant weakness, the manuscript presents some issues and needs more details and clarity concerning methodological elements that are useful for understanding its content.

Therefore, my overall peer review judgment is a major revision.

¶AREAS OF IMPROVEMENT

#GENERAL:

*abbreviations: please report ALL the abbreviations in full in the whole manuscript.

#TABLES and FIGURES:

*abbreviations: please report ALL the abbreviations in full in a legend.

#ABSTRACT:

*objective: “to systematically review” is better than “to create an overview”.

#INTRODUCTION

*background and rationale: The authors rightly point out that there is a lack of qualitative work in the thoracic spine. However, qualitative systematic reviews on neck pain and low back pain already exist to date. The authors point out that the quality of these is suboptimal using AMSTAR 2. I do not understand, how creating a review of the entire spine could increase the methodological quality since the unit of measurement is the same as in the previous reviews, i.e., the qualitative primary works. I invite the authors to clarify this point more because the rationale is not strong enough to justify a general work. Perhaps, it might make more sense to focus on the backbone that is lacking in synthesis. Thank you.

#METHODS:

*review team: was it multidisciplinary or only physios were involved?

*registration on prospero: was it prospective?

*search strategy: provide the full strategy for each database.

*inclusion/exclusion criteria: it is not clear if mixed methods will be included? I mean the qualitative section of mixed methods. Please explain.

*risk of bias: will you adopt any tools to assess the methodological quality in the mixed methods study? This Is not clear.

*meta-aggregation: I suggest authors be more balanced when presenting the pros and dons of other qualitative methods (e.g., meta-ethnography, metasummary).

7. PLOS authors have the option to publish the peer review history of their article (what does this mean?). If published, this will include your full peer review and any attached files.

Reviewer #1: No

Reviewer #2: No

---

## [Author Response · Author response to Decision Letter 0]

18 Jun 2024

Reviewer #1: 

*Generally, this qualitative systematic review can contribute significantly to the literature on MSK pain management, and the description of the methods is comprehensive. However, the authors need to flesh out the uniqueness, innovative contribution, or implication of the study. The potential impact or “so what” question is not well articulated yet. The allusion to patient-centeredness is generic and has been studied by similar reviews. A quick search shows several of them.

*One of your arguments on page 6 is that a comprehensive synthesis that includes neck, thoracic, and lower back regions is necessary. However, you did not provide a clear justification, in the paragraph, for that assertion. What would it add to the literature or its implication for practice?

Response: We appreciate the reviewer’s comments and have revised our manuscript to highlight the unique contributions and implications of our study more clearly as detailed below.

Uniqueness and Innovative Contribution:

Inclusion of Thoracic Pain: Our review uniquely includes thoracic pain, a region underrepresented in qualitative research (and quantitative), addressing a critical gap in the literature.

Comprehensive combined synthesis in addition to region-specific synthesis (the dual approach highlighted in the objectives): By synthesizing data from cervical, thoracic, and lumbar regions, we identify overarching themes and unique differences in patient experiences, providing a nuanced understanding of MSK pain.

Enhanced Methodological Rigor: Unlike other studies we use the JBI tool for quality assessment, offering a balanced evaluation of qualitative studies and reducing bias (and other items page 6-7)

Implications for Practice:

Informing Patient-Centered Care: Our findings could help tailor more effective, patient-centered care strategies by addressing unique and common challenges across spinal regions.

Guiding Integrated Care Strategies: The comprehensive synthesis could guide the development of integrated care strategies that can address the multifaceted nature of MSK pain, improving patient outcomes.

Addressing Research Gaps: Including thoracic pain and synthesizing data across all spinal regions fills significant gaps in the literature, spurring further research into underexplored areas of MSK pain.

Changes: revised manuscript on pages 6-7

*The correct meaning of CADTH is “Canadian Agency for Drugs and Technologies in Health” (p. 10, line 187-188).

Response: Thank you, we have corrected "CADTH" to "Canadian Agency for Drugs and Technologies in Health." (p.11)

*There seem to be some discordance between your sub-objective on quality decision/confidence which stems from your critique on the low quality of previous reviews (in page 6, lines 112-118) and your decision (in page 13, lines 249-252) not to exclude papers regardless of the quality rating. It is not very clear how attaching lower priority to the “lower quality” papers would steer the overall integrity of your review findings.

Response: As stated in the manuscript, our review aims to capture a broad spectrum of perspectives by including all relevant studies, regardless of quality, to avoid further narrowing an already limited pool of evidence and recognizing that each qualitative study participant can provide unique insights into patients’ experiences. Therefore, excluding studies based on quality thresholds might result in the loss of valuable information.

To address the influence of low-quality studies, we will assign a quality score (%), where higher quality studies will have more weight (higher plausibility level that is essential in meta-aggregation) and more influence on the synthesized findings. This ensures that our conclusions remain robust and reflective of the best available evidence. Additionally, we transparently report quality scores and assign the levels of plausibility, enhancing the credibility of our review. By considering multiple factors in assessing confidence levels and not just study quality, we ensure that the synthesized findings maintain a high level of confidence.

Changes: We have clarified in the manuscript how attaching lower priority to lower quality studies maintains the overall integrity of our review findings by giving greater consideration to higher quality studies while still capturing the full range of available perspectives. This approach ensures that while all relevant studies are included, the conclusions drawn are primarily informed by higher quality evidence, thus maintaining the robustness and credibility of our synthesized results. See pages 14-15.

*The manuscript needs proofreading before resubmission: punctuations and inconsistent font size

Response: We have thoroughly proofread the manuscript and corrected all punctuation, grammatical errors, and font size inconsistencies (Font sizes used: Body 12pt, Level 1 headings: 18pt, Level 2 headings: 16pt, Level 3 headings: 14pt).

Reviewer #2: 

¶AREAS OF IMPROVEMENT

#GENERAL:

*abbreviations: please report ALL the abbreviations in full in the whole manuscript.

Response: Thank you for your valuable feedback. We have ensured that all abbreviations are defined on first appearance in the manuscript in line with the PlosOne guidelines for authors.

#TABLES and FIGURES:

*abbreviations: please report ALL the abbreviations in full in a legend.

 Response: We have added a list of all abbreviations under the supplementary table.

#ABSTRACT:

*objective: “to systematically review” is better than “to create an overview”.

Response: We appreciate the reviewer’s feedback. We have changed the text from “to provide a comprehensive overview” to “to provide a comprehensive systematic review of the literature” (p.15). 

#INTRODUCTION

*background and rationale: The authors rightly point out that there is a lack of qualitative work in the thoracic spine. However, qualitative systematic reviews on neck pain and low back pain already exist to date. The authors point out that the quality of these is suboptimal using AMSTAR 2. I do not understand, how creating a review of the entire spine could increase the methodological quality since the unit of measurement is the same as in the previous reviews, i.e., the qualitative primary works. I invite the authors to clarify this point more because the rationale is not strong enough to justify a general work. Perhaps, it might make more sense to focus on the backbone that is lacking in synthesis. Thank you.

Response: We appreciate the reviewer’s comments and would like to clarify how our comprehensive review could improve on items of the AMSTAR 2 checklist:

Protocol Established Prior to Review (Item 2): Our methods were pre-established and registered.

Comprehensive Literature Search Strategy (Item 4): By expanding our search to include all spinal regions, we ensure a more thorough and inclusive literature search strategy, which addresses the need for a broader evidence base and reduces the risk of missing relevant studies. Our search constructs were informed by robust and established sources, includes multiple databases, grey literature, and consultation with experts, ensuring thoroughness.

Description of Included Studies (Item 8): Our review will provide a detailed description of the characteristics of included studies, including the context and specific spinal regions involved. This level of detail helps in understanding the applicability and transferability of the findings.

Risk of Bias Assessment (Item 9): Previous reviews used quality assessment tools biased toward certain paradigms or methodologies of qualitative research. In contrast, we use the JBI tool, which offers a more balanced and comprehensive assessment of qualitative studies, ensuring methodological quality and reducing bias.

By addressing these aspects, our comprehensive review aims to enhance the methodological quality and provide a more robust and holistic synthesis of MSK spinal pain experiences, improving upon the limitations of previous reviews.

Changes: revised manuscript on pages 6-7.

#METHODS:

*Review team: was it multidisciplinary or only physios were involved?

Response: only physios were involved.

Changes: added in design paragraph:” The review team consists of physical therapy researchers with expertise in qualitative research, systematic reviews and spinal pain. “

*Registration on prospero: was it prospective?

Response: It was prospective.

Changes: added in design section: “…and prospectively registered in PROSPERO (International prospective register of systematic reviews) to enhance …”

*Search strategy: provide the full strategy for each database.

Response: We appreciate the reviewer's suggestion to include the full search strategy. As this manuscript is a proposal, PRISMA P checklist states: “Present draft of search strategy to be used for at least one electronic database, including planned limits, such that it could be repeated”, The full search strategy will be provided in the published review. We will ensure that the detailed search strategy for each database is included in the final review manuscript as recommended by PRISMA statement.

Changes: added a note in the Methods - Search Strategy section: “The complete search strategy will be adapted for each database and will be included in the final review manuscript.” (p.11)

*inclusion/exclusion criteria: it is not clear if mixed methods will be included? I mean the qualitative section of mixed methods. Please explain.

*risk of bias: will you adopt any tools to assess the methodological quality in the mixed methods study? 

This Is not clear.

Response: We appreciate the reviewer's comment. As mentioned in the SPIDER tool section “…qualitative data from mixed methods research”, we will indeed include mixed methods studies. We are only interested in the qualitative aspects of these studies, and therefore, the quality assessment will be conducted using the same JBI tool specifically for the qualitative components as the quantitative parts of these studies will not be used or considered in our analysis. This approach is supported by methodologies outlined in several reviews where different components of mixed methods studies were assessed using the relevant JBI tool (quantitative and qualitative):

(- Silva TO, Ribeiro HG, Moreira-Almeida A. End-of-life experiences in the dying process: scoping and mixed-methods systematic review BMJ Supportive & Palliative Care 2023;13:e624-e640

- Atkins, S., Launiala, A., Kagaha, A. et al. Including mixed methods research in systematic reviews: Examples from qualitative syntheses in TB and malaria control. BMC Med Res Methodol 12, 62 (2012). https://doi.org/10.1186/1471-2288-12-62).

Changes: Added in Methods-Eligibility criteria section: “We will be including mixed methods studies, but we are specifically interested in their qualitative data, and the quantitative aspects will not be used or included in our analysis.” (p.9)

 Added in Quality assessment section: For mixed methods studies, quality assessment will focus solely on the qualitative components using the same JBI tool for qualitative studies, this ensures that we assess the methodological quality relevant to our review’s qualitative focus. (p.13)

*meta-aggregation: I suggest authors be more balanced when presenting the pros and dons of other qualitative methods (e.g., meta-ethnography, metasummary).

Response: We appreciate the reviewer's suggestion and have revised the Methods - Data Synthesis.

Changes: “We recognize and acknowledge the issues with synthesizing qualitative data, as researchers explained that some qualitative data synthesis methods like meta synthesis, allows for a deep, interpretative synthesis that can generate new theoretical insights and understanding, providing a rich and nuanced comprehension of the research phenomenon. However, it involves interpreting an interpretation which would risk losing the core insights of the original studies. This method often involves elevating data to a more abstract level by using analytical methods like those in the initial studies (e.g. thematic analysis) …” (p.15)

---

## [Decision Letter · Decision Letter 1]

26 Jun 2024

Patients’ Experiences with Musculoskeletal Spinal Pain: A Qualitative Systematic Review Protocol

PONE-D-24-07892R1

Dear Dr. El Chamaa,

We’re pleased to inform you that your manuscript has been judged scientifically suitable for publication and will be formally accepted for publication once it meets all outstanding technical requirements.

Kind regards,

Adedayo Ajidahun

Academic Editor

PLOS ONE

Additional Editor Comments (optional):

Reviewers' comments:

Reviewer's Responses to Questions

**Comments to the Author**

1. Does the manuscript provide a valid rationale for the proposed study, with clearly identified and justified research questions?

Reviewer #1: Yes

Reviewer #2: Yes

2. Is the protocol technically sound and planned in a manner that will lead to a meaningful outcome and allow testing the stated hypotheses?

Reviewer #1: Yes

Reviewer #2: Yes

3. Is the methodology feasible and described in sufficient detail to allow the work to be replicable?

Reviewer #1: Yes

Reviewer #2: Yes

4. Have the authors described where all data underlying the findings will be made available when the study is complete?

Reviewer #1: Yes

Reviewer #2: Yes

5. Is the manuscript presented in an intelligible fashion and written in standard English?

Reviewer #1: Yes

Reviewer #2: Yes

6. Review Comments to the Author

You may also provide optional suggestions and comments to authors that they might find helpful in planning their study.

Reviewer #1: Thank you for offering me the opportunity to review this manuscript. I am satisfied with the authors' responses and efforts to address the concerns I raised.

Reviewer #2: Thanks for your explanation and ameliorations. I am curious to read the final version of the review.

Best regards.

7. PLOS authors have the option to publish the peer review history of their article (what does this mean?). If published, this will include your full peer review and any attached files.

Reviewer #1: No

Reviewer #2: No

---

## [Editor Report · Acceptance letter]

1 Jul 2024

PONE-D-24-07892R1 

PLOS ONE

Dear Dr. El Chamaa, 

I'm pleased to inform you that your manuscript has been deemed suitable for publication in PLOS ONE. Congratulations! Your manuscript is now being handed over to our production team.

Kind regards, 

on behalf of

Dr. Adedayo Ajidahun 

Academic Editor

PLOS ONE